# PROOF ARTIFACT CO-TRAINING FOR THEOREM PROVING WITH LANGUAGE MODELS

**Jesse Michael Han**
University of Pittsburgh
OpenAI

**Jason Rute**
IBM Research*

**Yuhuai Wu**
Google Research
Stanford University†

**Edward W. Ayers**
Carnegie Mellon University‡

**Stanislas Polu**
OpenAI

## ABSTRACT

Labeled data for imitation learning of theorem proving in large libraries of formalized mathematics is scarce, as such libraries require years of concentrated effort by human specialists to be built. This is particularly challenging when applying large Transformer language models to tactic prediction, because the scaling of performance with respect to model size is quickly disrupted in the data-scarce, easily-overfitted regime. We propose PACT (**P**roof **A**rtifact **C**o-**T**raining), a general methodology for extracting abundant self-supervised data from kernel-level proof terms for joint training alongside the usual tactic prediction objective. We apply this methodology to Lean, a proof assistant host to some of the most sophisticated formalized mathematics to date. We instrument Lean with a neural theorem prover driven by a Transformer language model and show that PACT improves theorem proving success rate on a held-out suite of test theorems from 32% to 48%.

## 1 INTRODUCTION

Deep learning-driven automated theorem proving in large libraries of formalized mathematics (henceforth "neural theorem proving") has been the focus of increased attention in recent years. Labeled data for imitation learning of theorem proving is scarce—formalization is notoriously labor-intensive, with an estimated cost of 2.5 man-years per megabyte of formalized mathematics (Wiedijk, 2000), and complex projects require years of labor from human specialists. Within a fixed corpus of (possibly unproven) theorem statements, it is possible to augment a seed dataset of human proofs with new successful trajectories using reinforcement learning or expert iteration. However, for some large models this can be quite computationally intensive, and without a way to expand the curriculum of theorems, the agent will inevitably saturate and suffer from data starvation.

Data scarcity is a particularly thorny obstruction for applying large language models (LLMs) to neural theorem proving. LLMs have achieved spectacular success in data-rich regimes such as plain text (Brown et al., 2020), images (Dosovitskiy et al., 2021), and joint text-image modeling (Radford et al., 2021), and the performance of decoder-only Transformers has been empirically shown to obey scaling power laws in model and data size (Henighan et al., 2020). However, existing datasets of human proof steps for neural theorem proving are extremely small and exist at scales at which overfitting occurs extremely rapidly, disrupting the scaling of performance with respect to model size (Kaplan et al., 2020).

We make two contributions towards addressing the problem of data scarcity in the context of formal mathematics. First, we introduce PACT (**P**roof **A**rtifact **C**o-**T**raining), a general methodology for extracting self-supervised auxiliary tasks for jointly training a language model alongside a tactic prediction objective for interactive theorem proving. Second, we present LEANSTEP, a collection of

---

*Work performed while Jason Rute was at CIBO Technologies.
†Work performed while Yuhuai Wu was at University of Toronto.
‡Work performed while Edward W. Ayers was at University of Cambridge.

datasets and a machine learning environment for the Lean 3 theorem prover with support for PACT, supervised learning of tactic prediction, theorem proving evaluation, and reinforcement learning.

We train large language models on these data and demonstrate that PACT significantly improves theorem proving success rate on a held-out suite of test theorems, from 32% to 48%. We then embark on a careful study of the effects of pre-training vs. co-training and show that PACT combined with *WebMath* pre-training (Polu & Sutskever, 2020) achieves the best validation loss and theorem proving success rate. Finally, on an out-of-distribution collection of thousands of theorems (some involving novel definitions) added to Lean's mathematical library after we extracted our train/test data, we achieve a theorem proving success rate of 37%, suggesting strong generalization and usefulness at the frontier of formalized mathematics.

## 2 BACKGROUND AND RELATED WORK

LEAN    Lean is an interactive theorem prover and functional programming language (de Moura et al., 2015). It has an extremely active community and is host to some of the most sophisticated formalized mathematics in the world, including scheme theory (Buzzard et al., 2021), forcing (Han & van Doorn, 2020), perfectoid spaces (Buzzard et al., 2020), and condensed mathematics (Scholze, 2020). Lean's foundational logic is a dependent type theory called the calculus of inductive constructions (Pfenning & Paulin-Mohring, 1989). This design means that terms, types and proofs are all represented with a single datatype called an *expression*. A *proof term* is a Lean expression whose type is a proposition, *i.e.* a theorem. This proof term serves as a checkable artifact for verifying the proposition. Lean uses a small, trusted kernel to verify proof terms. The primary repository of formalized mathematics in Lean is `mathlib` (mathlib, 2020). At the time of writing, 140 contributors have added almost 500,000 lines of code; `mathlib` contains over 46,000 formalized lemmas backed by over 21,000 definitions, covering topics such as algebraic geometry, computability, measure theory, and category theory. The range of topics and the monolithic, unified organization of `mathlib` make it an excellent foundation for a neural theorem proving dataset.

MACHINE LEARNING IN INTERACTIVE THEOREM PROVING    In a tactic-based interactive theorem prover (ITP) such as Lean, a proof is a list of tactics, *i.e.* small proof-term-generating programs. Tactics can be simple one-word commands, *e.g.* `refl`, or be composed of many nested parts, *e.g.*

```
simpa [le_antisymm_iff, norm_nonneg] using @norm_eq_zero α _ g
```

Here the brackets enclose a list of simplifier rules (which often are just lemmas from the library), and `@norm_eq_zero α _ g` is a proof term applying the lemma `norm_eq_zero` to the local variables $\alpha$ and `g`.

Other ML and neural theorem provers for tactic-based ITPs take one of two approaches to tactic generation. TacticToe (Gauthier et al., 2021) for HOL4 and Tactician (Blaauwbroek et al., 2020) for Coq use k-NN to select similar tactics in the training set and apply modifications to the result, *e.g.* swapping the tactic variables with those found in the local context. HOList/DeepHOL (Bansal et al., 2019b;a; Paliwal et al., 2020) for HOL Light; TacticZero (Wu et al., 2021a) for HOL4; and CoqGym/ASTactic (Yang & Deng, 2019) and ProverBot9001 (Sanchez-Stern et al., 2020) for Coq hard-code the DSL for every tactic command. The model chooses a tactic command, and then fills in the tactic arguments using specialized argument selectors (such as a lemma selector, a local hypothesis selector, and/or a variable selector). None of these selectors currently synthesize arbitrary terms. This prevents the tactic synthesis from constructing tactics with proof terms, such as `@norm_eq_zero α _ g`, or directly proving an existential, *e.g.* $\exists$ `(x : ℝ), x + 3 = 0`, by supplying the witnessing term `-3`.

Directly applying generative language modeling to tactic generation allows this setup to be considerably simplified. Our tactic generator is able to synthesize tactics of any form found in `mathlib` including, for example, the `simpa` example above as a one line proof to a test theorem, even though the string `@norm_eq_zero` does not occur in our dataset. (See more examples in Appendix D.) We leave as future work the possibility of re-integrating specialized components, *e.g.* lemma selection, found in other works (possibly as, say, a source of additional prompts for the language model).

Language models have also been explored in the first-order ITP Mizar for conjecturing and proof synthesis (Urban & Jakubuv, 2020). While their work shows the promise of such approaches,

is not intended as a complete end-to-end theorem prover. For Metamath, which does not use tactics, language modeling approaches have been quite successful. Holophrasm (Whalen, 2016), MetaGen (Wang & Deng, 2020), and GPT-f (Polu & Sutskever, 2020) all use RNNs or Transformers to generate proof steps. Indeed, our paper builds on the work of Metamath GPT-f (Polu & Sutskever, 2020) (MM GPT-f). Whereas MM GPT-f trained primarily on the Metamath proof step objective (*i.e.* guessing the next lemma to be applied to a goal, which is similar to our NEXTLEMMA task in Section 3.2), we co-train on a diverse suite of self-supervised tasks extracted from Lean proof terms and demonstrate significant improvements in theorem proving performance when doing so. This is our main result.

REASONING WITH TRANSFORMERS    Besides theorem proving, a number of recent papers have shown that language models, especially Transformers, are capable of something like mathematical and logical reasoning in integration (Lample & Charton, 2020), differential equations (Charton et al., 2021), Boolean satisfiability (Hahn et al., 2021), and inferring missing proof steps (Li et al., 2021).

A closely-related vein of work has shown that pre-training Transformers on data engineered to reflect inductive biases conducive to mathematical reasoning is beneficial for downstream mathematical reasoning tasks (Rabe et al., 2021; Wu et al., 2021b). Our work both builds on and departs from these ideas in several ways. Unlike skip-tree training (Rabe et al., 2021), which focuses solely on predicting masked subterms of theorem *statements*, PACT derives its self-supervised training data from far more complex *proofs*. Unlike LIME (Wu et al., 2021b), which uses purely synthetic data and is presented as a pre-training methodology, our self-supervised tasks are extracted from non-synthetic human proofs. Moreover, we show that not only are Transformers capable of performing well on auxiliary tasks gathered from low-level proof artifact data, but that we can directly leverage this low-level data by jointly training a language model to greatly improve its performance at high-level theorem proving.

MACHINE LEARNING WITH PROOF ARTIFACTS    The idea of mining low-level proof artifacts was previously explored by Kaliszyk and Urban in the context of automated lemma extraction (Kaliszyk & Urban, 2015b; Kaliszyk et al., 2015). It has also been previously observed that training on fully elaborated Coq terms (Nie et al., 2020) helps with a downstream theorem naming task. However, similar to previous work on skip-tree training, their dataset focuses solely on theorem statements, *i.e.* types, does not cover the far more complex proof terms, and does not evaluate the effect of such training on theorem proving evaluations.

While there exist environments and datasets for other formal mathematics libraries (Kaliszyk et al., 2017; Li et al., 2021; Huang et al., 2019; Kaliszyk & Urban, 2015a), LEANSTEP is the first and only tactic proof dataset for the Lean theorem prover. This makes available a large set of formal mathematical data to researchers covering a diverse and deep spectrum of pure mathematics. Moreover, LEANSTEP is unique in that it contains both high-level human-written tactics as well as kernel-level proof terms, which enables the extraction of self-supervised tasks for PACT (Section 3.2).

## 3    THE LEANSTEP DATASETS AND MACHINE LEARNING ENVIRONMENT

### 3.1    HUMAN TACTIC PROOF STEPS

Tactics in Lean are metaprograms (Ebner et al., 2017), which can construct Lean expressions, such as proof terms. A *tactic state* which tracks the list of open goals and other metadata (like the partial proof term constructed so far) is threaded through each tactic invocation. Lean has special support for treating tactics as an extensible domain-specific language (DSL); this DSL is how Lean is typically used as an interactive theorem prover. The DSL amounts to a linear chain of comma-separated invocations. The Lean *proof step* task is to predict the next tactic given this goal state. We refer the reader to Appendix A for examples and further explanation.

Our human tactic proof step dataset consists of source-target pairs of strings, one for each tactic invocation in the Lean core library and in `mathlib`. The source string is the pretty-printed tactic state. The target string is the tactic invocation as entered by a human author of the source code. This data is gathered by hooking into the Lean parser and Lean's compilation process. We refer to the task of predicting the next human tactic proof step given a tactic state as the *proofstep objective*.

### 3.2 Proof artifact co-training

In this section, we describe the PACT task suite and how data for these tasks are extracted.

For every proof term $\tau$, we record the type $\Gamma$ of $\tau$, its name nm, and a list ps of all premises (*i.e.* named references to other lemmas in the library) which are used in $\tau$. We then recurse through $\tau$, tracking a list bs of bound variables which we update whenever navigating into the body of a $\lambda$-expression. At every sub-term $\tau' \subseteq \tau$ we record $\tau'$, its type $\Gamma'$, the current state of bs, and the following data:

1. A *tactic state*, where the goal is set to be $\Gamma'$ and the list of hypotheses in the local context is set to be the list bs, *i.e.* those bound variables in scope at $\tau'$.
2. A *partial proof term*, *i.e.* $\tau$ with $\tau'$ masked out.
3. A *premise selection bitmask*, *i.e.* Boolean labels for every p in ps indicating whether p is used in $\tau'$.
4. A *local context bitmask*, *i.e.* similar Boolean labels for every b in bs indicating whether b is used in $\tau'$.
5. An optional *next lemma*: if the first step of $\tau'$ is to apply a premise p in ps, we record p.

Whenever we record a term, we record both *pretty-printed* and far more explicit *fully elaborated* versions of it. The fully elaborated terms explicitly display enormous amounts of type information which are usually silently inferred by Lean. From these data, we assemble the following language modeling tasks:

1. **Next lemma prediction.** Given the tactic state, predict the next lemma to be applied.
2. **Proof term prediction.** Given the tactic state, predict the entire proof term $\tau'$.
3. **Skip-proof.** Given the partial proof term, predict the masked subterm $\tau'$.
4. **Type prediction.** Given the partial proof term, predict the type $\Gamma'$ of the masked subterm $\tau'$.
5. **Tactic state elaboration.** Given the tactic state, predict the fully elaborated tactic state.
6. **Proof term elaboration.** Given $\tau$, predict the fully elaborated version of $\tau$.
7. **Premise classification.** Given the tactic state and a premise $p \in ps$, predict either <TRUE> or <FALSE> according to the premise selection bitmask.
8. **Local context classification.** Given the tactic state (which consists of a list of local assumptions bs and the goal $\Gamma'$), predict the sublist of bs which is true on the local context bitmask.
9. **Theorem naming.** Given the type $\Gamma$ of the top-level proof term $\tau$, predict the name nm.

We remark that our next lemma prediction task is precisely the low-level PROOFSTEP objective studied in (Polu & Sutskever, 2020), and our skip-proof task superficially resembles, but is much more difficult than the skip-tree task studied in (Rabe et al., 2021), as proof terms tend to be far more complex than the syntax trees of theorem statements.

### 3.3 The LeanStep machine learning environment

We instrument Lean for automatic theorem proving with a language model, including utilities for (1) setting the runtime environment at a particular theorem (ensuring proofs are never circular), (2) serializing the tactic state as environment observations for a theorem-proving agent, (3) exposing Lean's parser to re-parse strings emitted by a language model into tactic invocations, and (4) executing and capturing the results of the re-parsed tactics, enabling the recording of trajectories for expert iteration and reinforcement learning.

In addition to this general instrumentation, we implement a generic best-first search algorithm for theorem proving; it forms the basis for our evaluations and is written entirely in Lean itself. The algorithm is parametrized by an oracle ($\Omega$ : tactic_state $\rightarrow$ list (string $\times$ float)) that accepts a tactic state and returns a list of strings and heuristic scores. The search is controlled

by a priority queue of *search nodes*, which consist of a tactic state (*i.e.* a partial proof) and search metadata. In the outer loop of the algorithm—which continues until either the theorem is completely proved (*i.e.* no goals are remaining on the current node), the priority queue is empty (*i.e.* the search has failed), or a pre-set timeout or budget of iterations is exceeded—we pop a node off the queue, serialize the associated tactic state and use it to query the oracle, producing a list of candidates `cs : list (string × float)`. We then loop over the candidates `cs` to produce a list of new search nodes, by re-parsing each string into a tactic and adding a new node if the parsed tactic advances the proof without raising errors. These new search nodes are then re-inserted into the queue in order of decreasing priority and the search continues. We optionally constrain the search by enforcing maximum width and depth limits $w_{\max}$ and $d_{\max}$ that guard insertion into the queue. When considering nodes for insertion, any node whose depth exceeds $d_{\max}$ is ignored, and all nodes are ignored if the queue size is strictly larger than $w_{\max}$. Due to the flexibility in assigning heuristic scores and in choosing the maximum width and depth hyperparameters, our algorithm is quite general—for example, it reduces to (1) a greedy depth-first search when $w_{\max} = 0$, and (2) a naïve breadth-first search when heuristic scores are identical and $w_{\max} = d_{\max} = \infty$.

## 4 EXPERIMENTS

TRAINING    In all of our experiments, we use decoder-only Transformers similar to GPT-3 (Brown et al., 2020). Unless mentioned otherwise, all of our models have 24 layers with $d_{\mathrm{model}} = 1536$ and 24 heads, accruing to 837M trainable parameters. They are also pre-trained on `WebMath` (Polu & Sutskever, 2020) for 72B tokens. We use the standard `BPE` encoding (Brown et al., 2020), a batch size of 512 and a learning rate of 0.00025 with a cosine schedule and a 100-step ramp-up.

We use an 80-5-15 train-validation-test split. We split all datapoints deterministically by *theorem name*, by hashing each name to a float in $(0, 1)$. This ensures, for example, that proof steps used to prove a test theorem never appear in the training data and vice-versa.

When fine-tuning a model we load its saved parameters but re-initialize the optimizer. We start each training for a fixed number of tokens (defining the cosine schedule) and record the number of tokens consumed as we reach a minimal validation loss. We use the minimum validation loss snapshot to evaluate each model on our held-out test set.

We partition our datasets into three groups:

1. `tactic`: the dataset described in Section 3.1.
2. `mix1`: the union of the PACT tasks **next lemma prediction** and **proof term prediction** (Section 3.2), selected because of their close relation to `tactic`.
3. `mix2`: all other datasets described in Section 3.2.

This grouping is motivated by the impossibility to ablate each dataset separately given our compute budget. They nonetheless enable us to study the effect of tasks that are very close to the `tactic` objective in comparison to others. Our choice of **next lemma prediction** and **proof term prediction** for `mix1` is motivated by the observation that these tasks are closely related to the theorem proving objective: a proof can be given entirely in terms of a sequence of lemmas to apply (as in Metamath), or the proof can be finished in one step by supplying the entire proof term. Despite their logical similarity to the `PROOFSTEP` objective, we nevertheless use different keywords in the prompt to the model to disambiguate (`NEXTLEMMA` and `PROOFTERM`) from (`PROOFSTEP`) because the data is noisy and represents a significant distribution shift: during pretty-printing, subtrees of proof terms beyond a certain depth are dropped entirely, there is generally no guarantee that they can be re-parsed, and the data is much more verbose than what humans typically supply in source code.

THEOREM PROVING EVALUATION    We run theorem-proving evaluations on our held-out `test` set, comprising 3071 theorems. Since the split was conducted by theorem name, the proofs of these theorems never appear in the training data. For each theorem in the test set, we set the runtime environment to the location where the theorem is proved in the source code, preventing the use of theorems defined later in `mathlib` and ensuring that we never derive circular proofs. We compare against existing proof automation In Lean by also evaluating the tactics `refl`, which attempts to prove statements via definitional equality, and `tidy`, which conducts a greedy depth-first search

```
tactic
```
| **tactic proof steps** | `GOAL <TacticState> PROOFSTEP <Tactic>` |
| --- | --- |

```
mix1
```
| **next lemma prediction** | `GOAL <TacticState> NEXTLEMMA apply (<NextLemma>)` |
| --- | --- |
| **proof term prediction** | `GOAL <TacticState> PROOFTERM exact (<ProofTerm>)` |

```
mix2
```
| **skip proof** | `RESULT <MaskedProofTerm> SKIPPROOF <ProofTerm>` |
| --- | --- |
| **type prediction** | `RESULT <MaskedProofTerm> PREDICTTYPE <Type>` |
| **tactic state elaboration** | `GOAL <TacticState> ELABGOAL <ElaboratedTacticState>` |
| **proof term elaboration** | `PROOFTERM <ProofTerm> ELABPROOFTERM <ElaboratedProofTerm>` |
| **premise classification** | `GOAL <TacticState> CLASSIFYPREMISE <Premise> <True\|False>` |
| **local context classification** | `GOAL <TacticState> CLASSIFYLOCALS <LocalsList>` |
| **theorem naming** | `TYPE <Type> NAME <Name>` |

Figure 1: Auto-regressive objectives used for each task described in Section 3. Placeholders represented with brackets (such as `<TacticState>`) are substituted by the context-completion pairs from each datasets in the prompts above. Each task is presented to the model with its respective keyword (`PROOFSTEP`, `NEXTLEMMA`,...). We wrap the completions of `mix1` tasks (with `apply(...)` and `exact(...)` respectively) as a hint that they are related to the respective Lean tactics; this is not directly possible for the other tasks.

using a fixed list of tactics at each step. We re-implement `tidy` as a special case of our best-first search algorithm using an oracle which always emits the same list of tactics, and so henceforth refer to it as `tidy-bfs`. In all of our experiments, we use a maximum width of $w_{max} = 16$, a maximum depth of $d_{max} = 128$, a maximum budget of $512$ iterations of the outer loop, a timeout of $5$ seconds per tactic execution, and a global timeout of $600$ seconds per theorem. Because sampling completions from our models is much slower ($\approx 1$ second) than querying the constant `tidy-bfs` oracle (instantaneous), the `tidy-bfs` search runs many more iterations than `gptf` before timeout.

We report the pass-rate (*i.e.* percentage of theorems proved) from the randomly-chosen held-out test set, following (Whalen, 2016), (Bansal et al., 2019c), and others. We provide an alternative pass-rate at the end of this section, using theorems added to `mathlib` after our dataset was collected. We average over three evaluation runs when reporting the pass rate.

EFFECT OF CO-TRAINING VS PRE-TRAINING   We first study the effects of pre-training versus co-training with the `mix1` and `mix2` datasets. We pre-train using the methodology described above (potentially pre-training first on `WebMath`, and then on a PACT dataset in sequence). For co-training, we simply concatenate and shuffle the datasets together without applying any particular weight to a given dataset.

The main results are presented in Figure 2. Pre-training exhibits an effective transfer from `mix-1` and/or `mix-2` but the best result is achieved by co-training with both these datasets. With this setup, we are able to train for much longer (71B tokens vs 22B+18B for the best pre-training setup) before overfitting on the `PROOFSTEP` task. We hypothesize that PACT regularizes overfitting to the `PROOFSTEP` task while still imparting useful knowledge to the model due to large amounts of mutual information, and that this is the main driver of increased performance.

ABLATING WEBMATH PRE-TRAINING   Next, we ablate the effect of `WebMath` pre-training (instead starting with a model pre-trained on the same English language mix as GPT-3). As expected, co-trained models suffer from a performance drop without `Webmath` pretraining. but we were more interested in measuring the effect on pre-trained models on `mix-1` and `mix-2`, as they may not benefit from `WebMath` as much due to the two successive pre-training steps.

We report the optimal validation losses in Figure 3. `WebMath` appears as substantially beneficial even in the sequential pre-training setup. This indicates that PACT is not a replacement for `WebMath` pre-training, but rather a complementary method for enhancing the performance of language models for theorem proving.

| Model | Tokens elapsed | mix1 | mix2 | tactic | Pass-rate |
|---|---|---|---|---|---|
| *Baselines* | | | | | |
| refl | | | | | 1.1% |
| tidy-bfs | | | | | 9.9% |
| WebMath > tactic | 1B | | | 1.02 | 32.2% |
| *Pre-training* | | | | | |
| WebMath > mix1 | 11B | *0.08* | | | |
| WebMath > mix2 | 16B | | *0.08* | | |
| WebMath > mix1 + mix2 | 22B | *0.11* | *0.08* | | |
| WebMath > mix1 > tactic | 1B | | | 1.00 | 39.8% |
| WebMath > mix1 + mix2 > tactic | 1B | | | 0.97 | 44.0% |
| *Co-training* (PACT) | | | | | |
| WebMath > mix1 + tactic | 18B | *0.08* | | 0.94 | 40.0% |
| WebMath > mix2 + tactic | 75B | | *0.09* | 0.93 | 46.0% |
| WebMath > mix1 + mix2 + tactic | 71B | *0.09* | *0.09* | **0.91** | **48.4**% |
| *Pre-training and co-training* | | | | | |
| WebMath > mix2 > mix1 + tactic | 18B | *0.08* | | 0.93 | 46.9% |

Figure 2: Comparison of pre-training and co-training on mix-1 and mix-2. > denotes a pre-training step and + denotes a co-training. As an example, WebMath > mix2 > mix1 + tactic signifies a model successively pre-trained on WebMath then mix2 and finally co-trained as a fine-tuning step on mix1 and tactic. Columns mix1, mix2, tactic report the optimal validation loss achieved on these respective datasets. We provide a detailed description of experiment runtime and computing infrastructure in Appendix B.

| Model | Tokens budget | Tokens elapsed | mix1 | mix2 | tactic | Pass-rate[†] |
|---|---|---|---|---|---|---|
| *Baselines* | | | | | | |
| tactic | 32B | 1B | | | 1.59 | — |
| *Pre-training* | | | | | | |
| mix1 | 32B | 20B | *0.12* | | | |
| mix2 | 32B | 25B | | *0.10* | | |
| mix1 + mix2 | 32B | 27B | *0.13* | *0.10* | | |
| mix1 > tactic | 32B | 1B | | | 1.26 | — |
| mix1 + mix2 > tactic | 32B | 1B | | | 1.16 | — |
| *Co-training* | | | | | | |
| mix1 + tactic | 32B | 27B | *0.11* | | 1.12 | — |
| mix2 + tactic | 96B | 75B | | *0.10* | **1.02** | 40.4% |
| mix1 + mix2 + tactic | 96B | 71B | *0.10* | *0.11* | 1.07 | — |
| *Pre-training and co-training* | | | | | | |
| mix2 > mix1 + tactic | 32B | 26B | *0.11* | | 1.09 | — |

Figure 3: Validation losses achieved in the pre-training and co-training setups without WebMath pre-training. See Figure 2 for a description of the columns and the models nomenclature used. [†]Due to technical constraints, we are unable to provide pass-rates for some of the models.

ABLATING REGULARIZATION We rule out the possibility that the benefits from PACT come from simply regularizing our models on the scarce tactic data alone. We checked that a WebMath > tactic model trained with 15% residual dropout achieved a minimum validation loss of 1.01 and 33.6% pass rate, far below the 48.4% PACT pass rate.

| Model | Tokens budget | Tokens elapsed | mix1 | mix2 | tactic | Pass-rate |
|---|---|---|---|---|---|---|
| 121m | 96B | 82B | *0.13* | *0.10* | 1.23 | 35.1% |
| 163m | 96B | 80B | *0.12* | *0.09* | 1.11 | 39.8% |
| 837m | 96B | 71B | *0.09* | *0.09* | **0.91** | **48.4%** |

Figure 4: Validation losses and pass-rates achieved for various model sizes using PACT. See Figure 2 for a description of the columns. The setup used is `WebMath > mix1 + mix2 + tactic`.

EFFECT OF MODEL SIZE    Finally, we study how performance scales with respect to model size. We use the best training setup reported in Figure 2, `WebMath > mix1 + mix2 + tactic`. The `837m` model is our main model. The `163m` and `121m` models respectively have 12 and 6 layers, with $d_{\mathrm{model}} = 768$. The learning rates are respectively adjusted to 0.0014 and 0.0016.

As demonstrated by Figure 4, performance is highly correlated with model size, with larger models generally achieving better generalization even in the overfitted regime. We leave as future work a careful study of how evaluation performance is affected when scaling to multi-billion parameter models, as well as the feasibility of deploying them for interactive use by Lean users.

TIME-STRATIFIED EVALUATION    In the 5 week period that separated our last dataset extraction and the writing of this paper, `mathlib` grew by 30K lines of code, adding 2807 new theorems. Evaluating our models on these new theorem statements gives a unique way to assess their capability to assist humans in formalizing proofs and to test their generalization to completely unseen theorems and definitions. This evaluation set also addresses one of the weaknesses of using a random split of theorems from a formal mathematics library, namely that the split is non-chronological; *e.g.* test theorems can appear as lemmas in proofs of train theorems.

We call this temporally held-out test set `future-mathlib` and evaluate our best model as well as the `refl` and `tidy-bfs` baselines on it. In contrast to evaluation on our `test` split, the `refl` baseline (simply attempting a proof by the `refl` tactic) closes 328 proofs (11.6%), demonstrating an important skew towards trivial boilerplate lemmas generally defined to provide alternate interfaces to new definitions. The `tidy-bfs` baseline closed 611 proofs (21.8%), and our best model `wm-tt-m1-m2` closed 1043 proofs (37.1%), proving 94% of the `refl` lemmas. We attribute the weaker performance to heavy distribution shift: by the nature of the dataset, the `future-mathlib` theorems frequently involve new definitions and concepts which the model was never exposed to during training. Nevertheless, the success rate remains high enough to suggest strong generalization and usefulness at the frontier of formalized mathematics.

## 5 DISCUSSION

CHAINED TACTIC PREDICTIONS    In Lean, multiple tactic commands can be chained together using semicolons. Our data pipeline treats these tactic chains as a single sequence in our training data, and they are occasionally predicted by the model. Such chained tactic applications are difficult for human formalizers to synthesize on their own, as they require reasoning about the semantics of multiple tactics in sequence and their effects on the tactic state, and the examples present in the training data are usually optimized by hand from longer, less succinct proofs. We observed that PACT significantly boosts the capability of our models to *successfully* predict longer chained tactic applications. This occurs despite the fact that the tactic chaining idiom is specific to the tactic proofstep dataset and does not appear in the PACT data whatsoever. We supply more detail in Appendix C.1.

THEOREM NAMING    We also evaluate our best PACT model (`wm-to-tt-m1-m2`) on the theorem naming task, using the theorem statements and human-supplied names from the `future-mathlib` evaluation set. It achieved 20% acc@1, 27% acc@10, and 30% acc@16. An inspection of its outputs reveals that even when its predictions diverge from the ground truth, they are often idiomatic and semantically correct alternatives. We supply more detail in Appendix C.2.

IMPACT ON LEAN COMMUNITY    Lean's `mathlib` (mathlib, 2020) is a rapidly growing open source library of formal mathematics which has grown considerably in size each year for the past four years.[1] Our work has been welcomed by members of this community, with Lean power users describing some of the new proofs found by GPT-f as "nontrivial" and "clever". More than one-third of the proofs found by our models are shorter and produce smaller proof terms (sometimes by several orders of magnitude) than the ground truth. Manually inspecting a small, non-cherry picked sample of these shorter proofs has led to 19 GPT-f co-authored commits to `mathlib`, some of which reduce proof term sizes and theorem compilation times by an order of magnitude (see Appendix D).

POTENTIAL SOCIETAL IMPACT    Strong automated reasoning systems have enormous potential impact for mathematical research and scientific progress in other disciplines. The methods that we discuss in this paper could accelerate the development of strong automated reasoning systems. We have also observed that our language models absorb stylistic biases from their training data which could be amplified via reinforcement learning. However, since we focus on mathematics codified in proof assistants, we believe that there is little immediate negative societal impact from our work.

FUTURE DIRECTIONS    There are many elaborations on the training data, training methodology, and tree search wrapping `lean-gptf` which can be reasonably expected to improve its performance at theorem proving. Our dataset can be synthetically augmented using similar methods as (Polu & Sutskever, 2020). Our dataset could be cleaned further, and proofs minimized. Merely making the decoded rewrites robust by only using the largest prefix of successful rewrites significantly boosts the success rate of suggested rewrites. In a similar vein, predicted lemmas generated as arguments to unsuccessful tactic applications could be cached and re-used as hints for an intermittently-queried hammer. The increased success rate of chained tactic predictions mentioned above shows the feasibility of having language models perform multiple reasoning steps in a single query, potentially improving the efficiency of the proof search. From the experiments described in Section 4, it is clear that the composition of the dataset used for co-training significantly affects performance on theorem proving. Although we uniformly sampled across all co-training tasks, it would be interesting to optimize a dynamic mixture schedule, perhaps annealing towards a desired task.

CONCLUSION    There is a sense in which PACT is merely an application of the well known principle that compute in the form of search should be exchanged for training signal whenever possible. In Lean, typeclass inference relies on a backtracking Prolog-style search; the elaborator performs search to disambiguate overloaded notation and infer types; Lean tactics have complex semantics precisely because they can perform search to find subproofs automatically. The work done by these subroutines is preserved in the proof artifacts, and PACT can be viewed as a way of extracting this information offline for more training signal.

We have presented PACT as a way of addressing the data scarcity issue for learning theorem proving from human tactic scripts in proof assistant libraries. Another well-studied solution for this is expert iteration and reinforcement learning. In the setting of HOL Light, and under the assumption of a hardcoded finite action space of tactics, Bansal et al. (2019a) in conjunction with supervised seed data was able to achieve up to 70% proof success rate on the HOList theorem proving task. Similarly, in a set-up much closer to ours, MM GPT-f demonstrated the feasibility of expert iteration when using generative language models for theorem proving.

Within a fixed corpus of theorems (and hence proof terms), however, both PACT and RL are fundamentally constrained by a lack of exploration—as the performance of the theorem proving agent improves, it will eventually saturate and become starved for data, and its curriculum will need to be expanded. Although self-supervised methods such as PACT represent a way to significantly improve the data-efficiency of reinforcement learning loops over existing theorem prover libraries, the development of continuously self-improving and infinitely scalable neural theorem provers remains contingent on sufficiently powerful exploration and automated curriculum generation; we consider these challenges to be of paramount importance.

---

[1]See `https://leanprover-community.github.io/mathlib_stats.html` for up-to-date statistics on `mathlib`'s size and growth over time.

## 6    ACKNOWLEDGMENTS

We thank the members of the Lean community, in particular Kevin Buzzard, Simon Hudon, Johan Commelin, Mario Carneiro, Bhavik Mehta, and Gabriel Ebner for their valuable feedback on our work. We are indebted to Markus Rabe and Christian Szegedy for many hours of helpful discussion. We also thank Daniel Selsam, Tom Hales, and Josef Urban for feedback on earlier drafts of this paper.

## 7    REPRODUCIBILITY STATEMENT

The source code used to generate the Lean datasets and run the evaluation is open source and made available in the following repositories:

**Lean theorem proving environment** :
> `https://github.com/jesse-michael-han/lean-tpe-public`

**Tactic step data pipeline** :
> `https://github.com/jasonrute/lean_proof_recording`

**PACT data pipeline** :
> `https://github.com/jesse-michael-han/lean-step-public`

Our Transformer model was pre-trained on two proprietary datasets. The first is the same mix used by GPT-3 (Brown et al., 2020) and the second is `WebMath` (Polu & Sutskever, 2020). More details can be found in Appendix B.

While our weights and the API through which we query our models are not currently public, techniques for training decoder-only transformers and efficiently performing inference with them are well-known. Our released theorem proving code is agnostic to these implementation details and will work with any language model exposed via an HTTP server. The provided code also supports querying a locally hosted Transformer from the open-source library `fairseq` via the Fairseq CLI (Ott et al., 2019).

We have released a simplified version of the proof search described in Section 3.3 as a tactic to the Lean community in a public beta, opening the way for our models to directly accelerate the development of formalized mathematics and for human experts to provide feedback and additional training signal in a virtuous cycle. The tactic and code are available at `https://github.com/jesse-michael-han/lean-gptf`, and users who sign up for the beta are granted access to our Transformer model through an API.

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

## A  ADDITIONAL BACKGROUND

PROOF TERMS  Lean's fundamental logic is a dependent type theory called the calculus of inductive constructions Pfenning & Paulin-Mohring (1989). This design means that terms ($4$, $x + y$, $f$), types ($\mathbb{N}$, `list` $\mathbb{Z}$, $\alpha \rightarrow \beta$) and proofs are all represented with a single datatype called an *expression*. Given an environment of available constants and definitions and a context $\Gamma$ of variables, Lean can infer a type $\alpha$ for each well-formed expression $t$. A *proof term* is a Lean expression whose type is a proposition. This proof term serves as a checkable artifact for verifying the proposition. Lean uses a small, trusted kernel to verify proof terms.

TACTICS  Tactics in Lean are metaprograms Ebner et al. (2017), which can construct Lean expressions, such as terms. A *tactic state* which tracks the list of open goals and other metadata is threaded through each tactic invocation. Lean has special support for treating tactics as an extensible domain-specific language (DSL); this DSL is how Lean is typically used as an interactive theorem prover. The DSL amounts to a linear chain of comma-separated invocations. The process of interactive proving is mediated through Lean's language server, which will present the context and type for the current goal in the proof to the user, dependent on where their cursor is in the source text. The *tactic prediction* task is to predict the next tactic given this goal state. We extract supervised training data for this task by extracting all human-supplied proof steps from Lean's `mathlib`.

An object called the *tactic state* is threaded through each invocation of a tactic. Among other things, the tactic state maintains a context of metavariables: placeholders in to which expressions will be

substituted later. At each point in the proof, one or more of these metavariables are selected as the *goal* of the tactic state which is present As the proof progresses, there are multiple values to be found

EXAMPLE    Consider this (modified) example of a tactic proof from the library.

```
theorem int.sub_ne_zero_of_ne : ∀ (a b : ℤ), a ≠ b -> a - b ≠ 0 :=
begin
    intros a b h hab,
    apply h,
    apply int.eq_of_sub_eq_zero hab,
end
```

Each tactic line modifies the proof state, which we explicitly annotate below with comments between each tactic.

```
theorem int.sub_ne_zero_of_ne : ∀ (a b : ℤ), a ≠ b -> a - b ≠ 0 :=
begin
    -- ⊢ ∀ (a b : ℤ), a ≠ b → a - b ≠ 0
    intros a b h hab,
    -- a b : ℤ,
    -- h : a ≠ b,
    -- hab : a - b = 0
    -- ⊢ false
    apply h,
    -- a b : ℤ,
    -- h : a ≠ b,
    -- hab : a - b = 0
    -- ⊢ a = b
    apply int.eq_of_sub_eq_zero hab,
    -- no goals
end
```

Our proofstep objective is to predict the tactic applied to a given tactic state.

Lean stores this proof internally as a proof term:

```
theorem int.sub_ne_zero_of_ne : ∀ (a b : ℤ), a ≠ b → a - b ≠ 0 :=
λ (a b : ℤ) (h : a ≠ b), id (λ (hab : a - b = 0), h
    (int.eq_of_sub_eq_zero hab))
```

Since this proof term is just stored internally as a tree, any branch of this term tree can be removed, to create a hole _, for example:

```
λ (a b : ℤ) (h : a ≠ b), id (λ (hab : a - b = 0), h  _)
```

Lean will automatically provide a list of both the local context and the type of a term needed to fill that hole as shown below. Notice this is the same as a tactic state we saw from the term proof above.

```
a b : ℤ,
h : a ≠ b,
hab : a - b = 0
⊢ a = b
```

Using this methodology of following proof term trees, we can mine low level proof data for every node of a term proof to produce the PACT dataset described in Section 3.2.

## B   DATASETS

### B.1   PRE-TRAINING DATASETS

We pre-train on `WebMath` as described in (Polu & Sutskever, 2020). All models, including the `WebMath` pre-trained models, and the non-`WebMath` models used in ablations, were first pre-trained on the mix used by GPT-3 (Brown et al., 2020) which includes a filtered `CommonCrawl`,

`WebText2`, `Book1`, `Book2` and `Wikipedia`. `WebMath` includes Python-only `GitHub` data, as well as `arXiv` and `Math StackExchange`.

From these datasets, a potential risk for test-set contamination (presence of `mathlib`) exists for the crawled datasets, namely `CommonCrawl`, `WebText2`, and (in case of a filtering bug) Python-only `GitHub`. The other datasets (in particular `arXiv` and `Math StackExchange`) may contain short references of `mathlib` code but in shape and forms that would not lead to effective contamination.

To assess the contamination risk related with the crawled datasets, we searched `CommonCrawl`, `WebText2`, `arXiv`, Python-only `GitHub`, and `Math StackExchange` for test theorems. For example, given the test theorem `nat.div_eq_sub_div` we searched for any occurrences of the string `div_eq_sub_div`. Of over 3000 test theorem names, we found 595 which occurred in the datasets. Many instances were innocuous, but some were in Lean files, and in some cases there was a proof of a test theorem. There were also 160 additional test theorems with no underscore in their name, which we did not check, but whose name is likely to be found in the datasets. (There is no need to check for training theorems since they are already in the training data and it would not constitute contamination.) We re-calculated the pass-rates of the results in Figure 2 omitting these 755 test theorems. This decreases the reported pass-rates slightly, ranging from 0.6 to 1.1 percentage points. The adjusted pass-rate of our best model `WebMath > mix1 + mix2 + tactic` is 47.4%, a decrease of 1 percentage point. Our main results still hold even with the adjusted pass-rates.

Additionally we also look at the results for the 1,350 test theorems in our dataset that were added to Lean and `mathlib` after April 18, 2020, which is after `CommonCrawl` and `WebText2` were gathered, and the 544 test theorems added after September 11, 2020, which is after `WebMath` was gathered. Unlike `future-mathlib`, these theorems were part of the originally extracted data. The pass-rates for the `WebMath > mix1 + mix2 + tactic` model on these restricted sets of test theorems are 45.6% and 43.3%, respectively.

We also looked for the following Metamath specific and HOL specific strings in `CommonCrawl`, `WebText2`, and Python-only `GitHub`:

```
Metamath:
    "( ph -> A = C )"
    "( ph -> A R C )"
    "( sqrt ` 2 ) e/ QQ"
HOL:
    "apply (rule "
    "apply (drule "
```

We found 0 occurrence of the Metamath-related strings but interestingly found a non-negligible amount of HOL-related documents, which does not constitute a test-set contamination but potentially benefits the downstream tasks studied in this paper.

While our results show a significant benefit to pre-training on `WebMath`, it is unclear exactly how pre-training helps. Since Lean's theorem names are made of coded mathematical phases, e.g. `affine.simplex.dist_circumcenter_eq_circumradius`, it is not unreasonable to suspect that important statistical connections are extracted from math sources. It is even possible that simple instances of auto-formalization or ITP translation are happening. There is prior work (Gauthier & Kaliszyk, 2015; Wang et al., 2018; 2020) suggesting that both of these are possible. From the point of view of a `lean-gptf` end-user, any such extraction of prior, publicly available data is useful and helpful. Nonetheless, our results are of a different nature than other AI for theorem proving research which do not use data outside of a given theorem proving library. This should be taken into account in any future comparisons and benchmarks.

## B.2 DATASET SIZES

- `tactic`: ≈128K examples.
- `mix1`
    - **Next lemma prediction**: ≈2.5M examples
    - **Proof term prediction**: ≈2.9M examples
- `mix2`

- **Skip-proof**: ≈1.7M examples
- **Type-prediction**: ≈1.7M examples
- **Tactic state elaboration**: ≈346K examples
- **Proof term elaboration**: ≈1.0M examples
- **Premise classification**: ≈9.3M examples
- **Local context classification**: ≈2.0M examples
- **Theorem naming**: ≈32$K$ examples.

## B.3 EXAMPLE DATAPOINTS

We present datapoints extracted from a toy example, namely the proof of the Peirce identity, viz.

```
lemma peirce_identity {P Q :Prop} : ((P → Q) → P) → P :=
begin
  apply or.elim (em P),
  intros h _,
  exact h,
  tauto!
end
```

From this, we can extract four `tactic` datapoints (i.e. human-generated tactic proof steps):

```
-- GOAL P Q : Prop ⊢ ((P → Q) → P) → P PROOFSTEP apply or.elim (em P)
-- GOAL P Q : Prop ⊢ P → ((P → Q) → P) → P  P Q : Prop ⊢ ¬P → ((P →
   Q) → P) → P PROOFSTEP intros h _
-- GOAL P Q : Prop, h : P, ǎ : (P → Q) → P ⊢ P  P Q : Prop ⊢ ¬P → ((P
   → Q) → P) → P PROOFSTEP exact h
-- GOAL P Q : Prop ⊢ ¬P → ((P → Q) → P) → P PROOFSTEP tauto!
```

In contrast, we can extract dozens of raw PACT datapoints. Due to space constraints, we list a representative sample of four such datapoints, from each of which we can derive the nine self-supervised auxiliary PACT tasks studied in our present work. For example, proof term prediction is precisely predicting the "proof_term" given the concatenation of "hyps", "⊢", and the "goal", skip-proof is predicting the "proof_term" given "result", etc.

```
  DATAPOINT:
---
{ "decl_nm":"peirce_identity",
  "decl_tp":"∀ {P Q : Prop}, ((P → Q) → P) → P",
  "hyps":[["P", "Prop"], ["Q", "Prop"], ["ǎ", "¬P"], ["ǎ_1", "(P → Q) →
    P"], ["ǎ_1", "¬(P → Q)"]],
  "hyps_mask":[true, false, false, false, false],
  "decl_premises":[["absurd", "∀ {a b : Prop}, a → ¬a → b"],
  ["absurd", "∀ {a b : Prop}, a → ¬a → b"],
  ["decidable.not_imp", "∀ {a b : Prop} [_inst_1 : decidable a], ¬(a →
   b) ↔ a ∧ ¬b"],
  ["iff.mp", "∀ {a b : Prop}, (a ↔ b) → a → b"],
  ["and.dcases_on",
   "∀ {a b : Prop} {C : a ∧ b → Prop} (n : a ∧ b), (∀ (left : a)
   (right : b), C _) → C n"],
  ["decidable.not_or_of_imp", "∀ {a b : Prop} [_inst_1 : decidable a],
   (a → b) → ¬a ∨ b"],
  ["or.dcases_on",
   "∀ {a b : Prop} {C : a ∨ b → Prop} (n : a ∨ b), (∀ (h : a), C _) →
   (∀ (h : b), C _) → C n"],
  ["em", "∀ (p : Prop), p ∨ ¬p"],
  ["or.elim", "∀ {a b c : Prop}, a ∨ b → (a → c) → (b → c) → c"]],
  "decl_premises_mask":[false, false, true, false, false, false, false,
   false, false],
  "goal":"∀ {b : Prop} [_inst_1 : decidable P], ¬(P → b) ↔ P ∧ ¬b",
  "proof_term":"decidable.not_imp",
```

```
  "result":"λ {P Q : Prop}, (em P).elim (λ (h : P) (ă : (P → Q) → P),
   h) (λ (ă : ¬P) (ă_1 : (P → Q) → P), (decidable.not_or_of_imp ă
   _1).dcases_on (λ (ă_1 : ¬(P → Q)), ((PREDICT Q
   (classical.prop_decidable P)).mp ă_1).dcases_on (λ (ă_1_left : P)
   (ă_1_right : ¬Q), absurd ă_1_left ă)) (λ (ă_1 : P), absurd ă_1 ă))",
  "next_lemma":["decidable.not_imp", "∀ {a b : Prop} [_inst_1 :
   decidable a], ¬(a → b) ↔ a ∧ ¬b"],
  "goal_is_prop":true,
  "verbose_proof_term":"@decidable.not_imp P",
  "verbose_goal":"∀ {b : Prop} [_inst_1 : decidable P], ¬(P → b) ↔ P ∧
   ¬b",
  "verbose_result":"λ {P Q : Prop}, (em P).elim (λ (h : P) (ă : (P → Q)
   → P), h) (λ (ă : ¬P) (ă_1 : (P → Q) → P),
   (@decidable.not_or_of_imp (P → Q) P (classical.prop_decidable (P →
   Q)) ă_1).dcases_on (λ (ă_1 : ¬(P → Q)), (@iff.mp (¬(P → Q)) (P ∧ ¬
   Q) (PREDICT Q (classical.prop_decidable P)) ă_1).dcases_on (λ
   (ă_1_left : P) (ă_1_right : ¬Q), @absurd P P ă_1_left ă)) (λ (ă_1 :
   P), @absurd P P ă_1 ă))"}
---

DATAPOINT:
---
{ "decl_nm":"peirce_identity",
  "decl_tp":"∀ {P Q : Prop}, ((P → Q) → P) → P",
  "hyps":[["P", "Prop"], ["Q", "Prop"], ["ă", "¬P"], ["ă_1", "(P → Q) →
   P"], ["ă_1", "¬(P → Q)"]],
  "hyps_mask":[false, true, false, false, false],
  "decl_premises":[["absurd", "∀ {a b : Prop}, a → ¬a → b"],
  ["absurd", "∀ {a b : Prop}, a → ¬a → b"],
  ["decidable.not_imp", "∀ {a b : Prop} [_inst_1 : decidable a], ¬(a →
   b) ↔ a ∧ ¬b"],
  ["iff.mp", "∀ {a b : Prop}, (a ↔ b) → a → b"],
  ["and.dcases_on",
   "∀ {a b : Prop} {C : a ∧ b → Prop} (n : a ∧ b), (∀ (left : a)
   (right : b), C _) → C n"],
  ["decidable.not_or_of_imp", "∀ {a b : Prop} [_inst_1 : decidable a],
   (a → b) → ¬a ∨ b"],
  ["or.dcases_on",
   "∀ {a b : Prop} {C : a ∨ b → Prop} (n : a ∨ b), (∀ (h : a), C _) →
   (∀ (h : b), C _) → C n"],
  ["em", "∀ (p : Prop), p ∨ ¬p"],
  ["or.elim", "∀ {a b c : Prop}, a ∨ b → (a → c) → (b → c) → c"]],
  "decl_premises_mask":[false, false, false, false, false, false, false,
   false, false],
  "goal":"Prop",
  "proof_term":"Q",
  "result":"λ {P Q : Prop}, (em P).elim (λ (h : P) (ă : (P → Q) → P),
   h) (λ (ă : ¬P) (ă_1 : (P → Q) → P), (decidable.not_or_of_imp ă
   _1).dcases_on (λ (ă_1 : ¬(P → Q)), (decidable.not_imp.mp ă
   _1).dcases_on (λ (ă_1_left : P) (ă_1_right : ¬Q), absurd ă_1_left ă
   )) (λ (ă_1 : P), absurd ă_1 ă))",
  "next_lemma":["Q", "Prop"],
  "goal_is_prop":false,
  "verbose_proof_term":"Q",
  "verbose_goal":"Prop",
  "verbose_result":"λ {P Q : Prop}, (em P).elim (λ (h : P) (ă : (P → Q)
   → P), h) (λ (ă : ¬P) (ă_1 : (P → Q) → P),
   (@decidable.not_or_of_imp (P → Q) P (classical.prop_decidable (P →
   Q)) ă_1).dcases_on (λ (ă_1 : ¬(P → Q)), ((@decidable.not_imp P
   PREDICT (classical.prop_decidable P)).mp ă_1).dcases_on (λ (ă_1_left
   : P) (ă_1_right : ¬Q), @absurd P P ă_1_left ă)) (λ (ă_1 : P),
   @absurd P P ă_1 ă))"}
---

DATAPOINT:
```

```
---
{ "decl_nm":"peirce_identity",
  "decl_tp":"∀ {P Q : Prop}, ((P → Q) → P) → P",
  "hyps":[["P", "Prop"], ["Q", "Prop"], ["ᾰ", "¬P"], ["ᾰ_1", "(P → Q) →
    P"], ["ᾰ_1", "¬(P → Q)"]],
  "hyps_mask":[true, true, false, false, false],
  "decl_premises":[["absurd", "∀ {a b : Prop}, a → ¬a → b"],
  ["absurd", "∀ {a b : Prop}, a → ¬a → b"],
  ["decidable.not_imp", "∀ {a b : Prop} [_inst_1 : decidable a], ¬(a →
  b) ↔ a ∧ ¬b"],
  ["iff.mp", "∀ {a b : Prop}, (a ↔ b) → a → b"],
  ["and.dcases_on",
   "∀ {a b : Prop} {C : a ∧ b → Prop} (n : a ∧ b), (∀ (left : a)
   (right : b), C _) → C n"],
  ["decidable.not_or_of_imp", "∀ {a b : Prop} [_inst_1 : decidable a],
  (a → b) → ¬a ∨ b"],
  ["or.dcases_on",
   "∀ {a b : Prop} {C : a ∨ b → Prop} (n : a ∨ b), (∀ (h : a), C _) →
   (∀ (h : b), C _) → C n"],
  ["em", "∀ (p : Prop), p ∨ ¬p"],
  ["or.elim", "∀ {a b c : Prop}, a ∨ b → (a → c) → (b → c) → c"]],
  "decl_premises_mask":[false, false, true, false, false, false, false,
   false, false],
  "goal":"∀ [_inst_1 : decidable P], ¬(P → Q) ↔ P ∧ ¬Q",
  "proof_term":"decidable.not_imp",
  "result":"λ {P Q : Prop}, (em P).elim (λ (h : P) (ᾰ : (P → Q) → P),
  h) (λ (ᾰ : ¬P) (ᾰ_1 : (P → Q) → P), (decidable.not_or_of_imp ᾰ
  _1).dcases_on (λ (ᾰ_1 : ¬(P → Q)), ((PREDICT
  (classical.prop_decidable P)).mp ᾰ_1).dcases_on (λ (ᾰ_1_left : P)
  (ᾰ_1_right : ¬Q), absurd ᾰ_1_left ᾰ)) (λ (ᾰ_1 : P), absurd ᾰ_1 ᾰ))",
  "next_lemma":["decidable.not_imp", "∀ {a b : Prop} [_inst_1 :
  decidable a], ¬(a → b) ↔ a ∧ ¬b"],
  "goal_is_prop":true,
  "verbose_proof_term":"@decidable.not_imp P Q",
  "verbose_goal":"∀ [_inst_1 : decidable P], ¬(P → Q) ↔ P ∧ ¬Q",
  "verbose_result":"λ {P Q : Prop}, (em P).elim (λ (h : P) (ᾰ : (P → Q)
   → P), h) (λ (ᾰ : ¬P) (ᾰ_1 : (P → Q) → P),
  (@decidable.not_or_of_imp (P → Q) P (classical.prop_decidable (P →
  Q)) ᾰ_1).dcases_on (λ (ᾰ_1 : ¬(P → Q)), (@iff.mp (¬(P → Q)) (P ∧ ¬
  Q) (PREDICT (classical.prop_decidable P)) ᾰ_1).dcases_on (λ
  (ᾰ_1_left : P) (ᾰ_1_right : ¬Q), @absurd P P ᾰ_1_left ᾰ)) (λ (ᾰ_1 :
  P), @absurd P P ᾰ_1 ᾰ))"}
---

DATAPOINT:
---
{ "decl_nm":"peirce_identity",
  "decl_tp":"∀ {P Q : Prop}, ((P → Q) → P) → P",
  "hyps":[["P", "Prop"], ["Q", "Prop"], ["ᾰ", "¬P"], ["ᾰ_1", "(P → Q) →
    P"], ["ᾰ_1", "¬(P → Q)"]],
  "hyps_mask":[false, false, false, false, false],
  "decl_premises":[["absurd", "∀ {a b : Prop}, a → ¬a → b"],
  ["absurd", "∀ {a b : Prop}, a → ¬a → b"],
  ["decidable.not_imp", "∀ {a b : Prop} [_inst_1 : decidable a], ¬(a →
  b) ↔ a ∧ ¬b"],
  ["iff.mp", "∀ {a b : Prop}, (a ↔ b) → a → b"],
  ["and.dcases_on",
   "∀ {a b : Prop} {C : a ∧ b → Prop} (n : a ∧ b), (∀ (left : a)
   (right : b), C _) → C n"],
  ["decidable.not_or_of_imp", "∀ {a b : Prop} [_inst_1 : decidable a],
  (a → b) → ¬a ∨ b"],
  ["or.dcases_on",
   "∀ {a b : Prop} {C : a ∨ b → Prop} (n : a ∨ b), (∀ (h : a), C _) →
   (∀ (h : b), C _) → C n"],
  ["em", "∀ (p : Prop), p ∨ ¬p"],
```

```
  ["or.elim", "∀ {a b c : Prop}, a ∨ b → (a → c) → (b → c) → c"]],
 "decl_premises_mask":[false, false, false, false, false, false, false,
  false, false],
 "goal":"Π (a : Prop), decidable a",
 "proof_term":"classical.prop_decidable",
 "result":"λ {P Q : Prop}, (em P).elim (λ (h : P) (ă : (P → Q) → P),
  h) (λ (ă : ¬P) (ă_1 : (P → Q) → P), (decidable.not_or_of_imp ă
  _1).dcases_on (λ (ă_1 : ¬(P → Q)), (decidable.not_imp.mp ă
  _1).dcases_on (λ (ă_1_left : P) (ă_1_right : ¬Q), absurd ă_1_left ă
  )) (λ (ă_1 : P), absurd ă_1 ă))",
 "next_lemma":["classical.prop_decidable", "Π (a : Prop), decidable a"],
 "goal_is_prop":false,
 "verbose_proof_term":"classical.prop_decidable",
 "verbose_goal":"Π (a : Prop), decidable a",
 "verbose_result":"λ {P Q : Prop}, (em P).elim (λ (h : P) (ă : (P → Q)
  → P), h) (λ (ă : ¬P) (ă_1 : (P → Q) → P),
  (@decidable.not_or_of_imp (P → Q) P (PREDICT (P → Q)) ă
  _1).dcases_on (λ (ă_1 : ¬(P → Q)), ((@decidable.not_imp P Q
  (PREDICT P)).mp ă_1).dcases_on (λ (ă_1_left : P) (ă_1_right : ¬Q),
  @absurd P P ă_1_left ă)) (λ (ă_1 : P), @absurd P P ă_1 ă))"}
---
```

## C EXPERIMENTS

### C.1 CHAINED TACTIC PREDICTION

Individual Lean tactics are chained together with commas. However, the Lean interactive tactic DSL also includes a number of other tactic combinators for creating composite tactics. A frequently used combinator is the infix semicolon `t; s` which will perform the tactic `t` and then apply the tactic `s` to each of the resulting subgoals produced by `t`. Our data pipeline for human tactic proof steps treats these semicolon-chained tactics as a single string for the language modeling objective. Thus, our models learn to occasionally emit multiple-step tactic predictions using semicolons. For example, `wm-to-tt-m1-m2` solved the following lemma in category theory with a single prediction chaining four tactics in a row:

```
theorem category_theory.grothendieck.congr
  {X Y : grothendieck F} {f g : X ⟶ Y} (h : f = g) :
  f.fiber = eq_to_hom (by subst h) ≫ g.fiber :=
begin
  rcases X; rcases Y; subst h; simp
end
```

One way of measuring the sophistication of predicted tactics is to consider the number of successful proofs on the evaluation set which have this composite form using semicolon-chaining. We display this analysis in Table 1, which shows that training with PACT in addition to the human-made tactics causes longer semicolon-chained tactics to be successfully predicted during theorem proving. This is remarkable because the semicolon idiom is specific to the tactic DSL which does not occur in the PACT data whatsoever, and yet the co-training causes longer and more frequent successful composite tactic predictions.

Table 1: Counting the number of semicolon-chained tactics predicted by our models that appear *in successful proofs*. Each column headed by a number $n;$ indicates the number of times that a suggestion appeared with $n$ occurrences of ';'.

| MODEL | 1; | 2; | 3; | 4; | MEAN |
|---|---|---|---|---|---|
| wm-to-tt | 215 | 49 | 2 | 0 | 1.199 |
| wm-to-tt-m1 | 186 | 39 | 5 | 1 | 1.225 |
| wm-to-tt-m1-m2 | **328** | **82** | **12** | **3** | **1.271** |

| | Correct top-1 guesses |
|---|---|
| **Theorem statement** | $\forall$ {$\alpha$ : Type u_1} {$\beta$ : Type u_2} [_inst_1 : decidable_eq $\alpha$] [_inst_2 : decidable_eq $\beta$] (s : finset $\alpha$) (t : finset $\beta$), s.product t = s.bUnion ($\lambda$ (a : $\alpha$), finset.image ($\lambda$ (b : $\beta$), (a, b)) t) |
| **Ground truth** | finset.product_eq_bUnion |
| **Theorem statement** | $\forall$ {$\alpha$ : Type u_1} {$\beta$ : Type u_2} [_inst_1 : topological_space $\alpha$] [_inst_2 : topological_space $\beta$] {f : $\alpha \to \beta$}, quotient_map f $\to$ function.surjective f |
| **Ground truth** | quotient_map.surjective |
| **Theorem statement** | $\forall$ {$\alpha$ : Type u_1} {$\beta$ : Type u_2} (f : $\alpha \to$ option $\beta$) (x : option $\alpha$), x.pbind ($\lambda$ (a : $\alpha$) (_x : a $\in$ x), f a) = x.bind f |
| **Ground truth** | option.pbind_eq_bind |
| **Theorem statement** | $\forall$ {C : Type u$_1$} [_inst_1 : category_theory.category C] {G : C $\Rightarrow$ C} [_inst_2 : category_theory.comonad G] {A B : category_theory.comonad.coalgebra G} (h : A.A $\cong$ B.A) (w : A.a $\gg$ G.map h.hom = h.hom $\gg$ B.a), (category_theory.comonad.coalgebra.iso_mk h w).hom.f = h.hom |
| **Ground truth** | category_theory.comonad.coalgebra.iso_mk_hom_f |
| **Theorem statement** | $\forall$ {**k** : Type u_1} {E : Type u_2} [_inst_1 : is_R_or_C ,**k**] [_inst_2 : inner_product_space **k** E] [_inst_4 : normed_space $\mathbb{R}$ E] [_inst_5 : is_scalar_tower $\mathbb{R}$ **k** E] (p x : E $\times$ E), $\Uparrow$(fderiv_inner_clm p) x = has_inner.inner p.fst x.snd + has_inner.inner x.fst p.snd |
| **Ground truth** | fderiv_inner_clm_apply |

Figure 5: A sample of correct top-1 guesses by our best model `wm-to-tt-m1-m2` on the *theorem naming* task. We performed this experiment on the `future-mathlib` evaluation set, which comprises entirely unseen theorems added to `mathlib` only after we last extracted training data.

### C.2 THEOREM NAMING CASE STUDY

We included *theorem naming* as part of the PACT task suite. By `mathlib` convention, theorem names are essentially snake-cased, natural language summaries of the type signature of a theorem, and so the theorem naming task is analogous to a formal-to-informal translation task. We evaluate the ability of our best model (in terms of theorem proving success rate) `wm-to-tt-m1-m2` on its ability to guess theorem names on the completely unseen `future-mathlib` set of theorems. The distribution shift inherent in the `future-mathlib` dataset particularly impacts the theorem naming task, because many of the ground-truth names will involve names for concepts that were only defined in `mathlib` *after* we extracted our training data.

On the $\approx$2.8K `future-mathlib` theorems, we queried `wm-to-tt-m1-m2` for up to $N = 16$ candidates. We order these candidates into a list `xs` by decreasing cumulative log-probability and calculate the top-$K$ accuracy by checking if any of the first $K$ candidates of `xs` match the ground

| | Incorrect guesses |
|---|---|
| **Theorem statement** | $\forall$ {$\alpha$ : Type u_1} (t : ordnode $\alpha$) (x : $\alpha$),
t.dual.find_min$'$ x = ordnode.find_max$'$ x t |
| **Guesses (top 8)** | ordinal.find_min$'$_eq, ordinal.find_min$'$_eq_max$'$, ordinal.find_min$'$_def,
ordinal.find_min$'$_eq_max, ordinal.find_min$'$, ordinal.dual_find_min$'$,
ordinal.find_min$'$_gt, ordinal.find_min$'$_q |
| **Ground truth** | ordnode.find_min$'$_dual |
| **Theorem statement** | $\forall$ {$\alpha$ : Type u_1} {$\beta$ : Type u_3} {$\gamma$ : Type u_5} [_inst_1 :
measurable_space $\alpha$] [_inst_3 : measurable_space $\beta$]
[_inst_5 : measurable_space $\gamma$] {$\mu$ : measure_theory.measure $\alpha$}
{$\nu$ : measure_theory.measure $\beta$}
[_inst_8 : measure_theory.sigma_finite $\nu$]
{f : $\alpha \times \beta \to \gamma$},
ae_measurable f ($\mu$.prod $\nu$) $\to$ ($\forall^{\mathrm{m}}$(x : $\alpha$) $\partial\mu$,
  ae_measurable ($\lambda$ (y : $\beta$), f (x, y)) $\nu$) |
| **Guesses (top 8)** | measure_theory.ae_prod, measure_theory.ae_of_ae_prod,
measure_theory.ae_eq_prod_of_ae, measure_theory.ae_ae_of_ae_prod,
measure_theory.ae_measure_prod_mk_left,
measure_theory.ae_prod_of_ae_prod,
measure_theory.ae_measure_prod, measure_theory.ae_eq_refl |
| **Ground truth** | ae_measurable.prod_mk_left |
| **Theorem statement** | $\forall$ {$\alpha$ : Type u_1} {$\beta$ : Type u_2} {$\gamma$ : Type u_3}
{f : filter $\alpha$} {h : set $\alpha \to$ set $\beta$} {m : $\gamma \to \beta$}
{l : filter $\gamma$}, filter.tendsto m l (f.lift$'$ h) $\leftrightarrow$
  $\forall$ (s : set $\alpha$), s $\in$ f $\to$ ($\forall^{\mathrm{f}}$ (a : $\gamma$) in l, m a $\in$ h s) |
| **Guesses (top 8)** | filter.tendsto_lift$'$_iff, filter.tendsto_lift$'$_def |
| **Ground truth** | filter.tendsto_lift$'$ |
| **Theorem statement** | $\forall$ {R : Type} [_inst_1 : comm_ring R]
{d : $\mathbb{Z}$} (f : $\mathbb{Z}\sqrt{}$d $\to$+$^*$ R),
$\uparrow$($\Uparrow$(zsqrtd.lift.symm) f) = $\Uparrow$f zsqrtd.sqrtd |
| **Guesses (top 8)** | zsqrtd.coe_lift_symm, zsqrtd.coe_lift.symm, zsqrtd.lift.coe_symm_apply,
zsqrtd.lift_symm_apply, zsqrtd.lift.coe_coe_symm,
  zsqrtd.lift.coe_symm_coe,
zsqrtd.lift.symm_coe_zsqrtd, zsqrtd.lift_symm_to_zsqrtd |
| **Ground truth** | zsqrtd.lift_symm_apply_coe |

Figure 6: A sample of incorrect guesses by our best model `wm-to-tt-m1-m2` on the *theorem naming* task. We performed this experiment on the `future-mathlib` evaluation set, which comprises entirely unseen theorems added to `mathlib` only after we last extracted training data. Most of the top-8 guesses displayed in the above table are very similar to the ground truth, in some cases being equivalent up to permutation of underscore-separated tokens. Note that for the first example, the concept of `ordnode` was not in the training data whatsoever and all predictions are in the syntactically similar `ordinal` namespace.

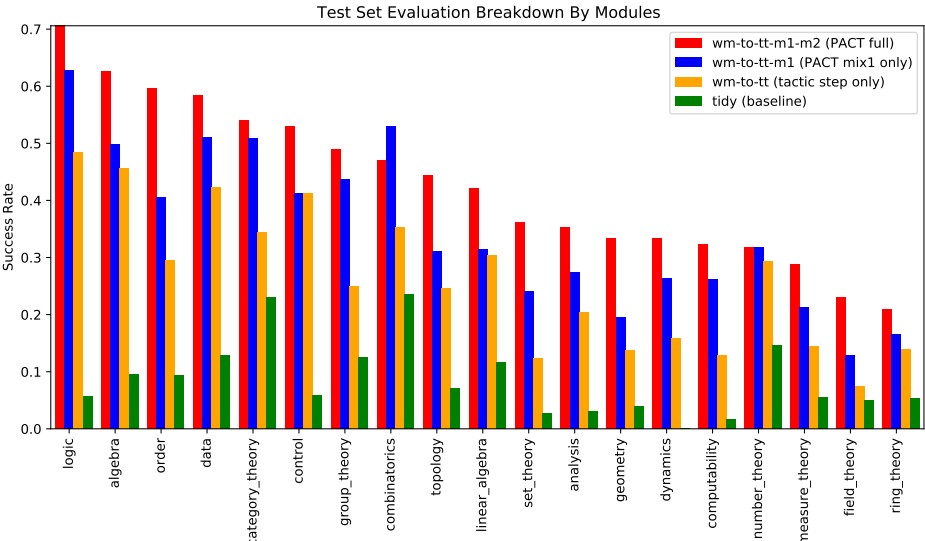

Figure 7: A breakdown of theorem proving success rate on the `test` set for `wm-to-tt-m1-m2`, `wm-to-tt-m1`, `wm-to-tt`, and the `tidy` baseline across top-level modules in Lean's `mathlib`. We see that `wm-to-tt-m1-m2` mostly dominates `wm-to-tt-m1` and the models trained using PACT dominate the model `wm-to-tt` trained on human tactic proof steps.

truth exactly. The model `wm-to-tt-m1-m2` was able to achieve 20.1% top-1 accuracy, 21.1% top-3 accuracy, 26.7% top-10 accuracy, and 30.0% top-16 accuracy. We display a sample of correct top-1 guesses (Figure 5) and a sample of failed guesses in (Figure 6). We note that the failed guesses, while containing no syntactic matches, are both semantically reasonable and syntactically very similar to the ground truth.

## C.3 TEST SET EVALUATION BREAKDOWN BY MODULE

Lean's `mathlib` is organized into top-level modules, which roughly organize theorems into mathematical subject area. In Figure 7, we break down the evaluation results on our `test` set between our PACT-trained models `wm-to-tt-m1-m2` and `wm-to-tt-m1` and our baselines `wm-to-tt` and `tidy`. We see that full PACT mostly dominates over co-training on just the `mix1` tasks over all subject areas, and that `wm-to-tt-m1` dominates the model `wm-to-tt` trained on human tactic proof steps only.

## C.4 BASELINE DESCRIPTION

The `tidy` backend is determined by a constant oracle

```
Ω : tactic_state → list (string × float)
```

which always returns the same list of tactics, namely:

```
meta def tidy_default_tactics : list (string × float) :=
list.map (flip prod.mk 0.0) [
    "refl"
,   "exact dec_trivial"
,   "assumption"
,   "tactic.intros1"
,   "tactic.auto_cases"
,   "apply_auto_param"
,   "dsimp at *"
,   "simp at *"
,   "ext1"
```

```
    ,   "fsplit"
    ,   "injections_and_clear"
    ,   "solve_by_elim"
    ,   "norm_cast"
]
```

Unlike the `gptf` backend, which generates a list of candidates in parallel independently, `tidy` enjoys the advantage that the list of tactics it emits is carefully chosen and ordered in order to optimize the proof search—this is based on the "waterfall" technique of the human-style automated theorem prover described in (Ganesalingam & Gowers, 2017).

### C.5 COMPUTATIONAL RESOURCE ESTIMATES

For each evaluation loop over the `test` set, we distributed the theorems over a pool of 32 CPU workers whose inference requests were load-balanced over 4 `V100` GPUs. Each evaluation required ≈10 hours with ≈30% GPU utilization. We observed that our evaluation was bottlenecked by inference and in practice, we hosted up to three evaluation loops at once on a VM with 80 logical cores without achieving full CPU utilization. In addition to the wall-clock timeout of 600s, we also limited the proof search to a logical timeout of 512 iterations, where one iteration corresponds to a single expansion of a node of the BFS search tree. In practice, so much time was spent either blocked on inference or performing the tactic executions in the inner loop of each iteration that we rarely exceeded the logical timeout, usually exceeding the wall-clock timeout instead.

Fine-tuning on our largest dataset `mix1 + mix2 + tactic` required 26 hours using 64 `A100` GPUs exhibiting high `FP16` usage, totalling an estimated ≈1.5K `A100(FP16)`-hours. This gives an estimated cost of 17.33 `A100(FP16)`-hours per billion elapsed tokens during training. We note that when calculating the number of elapsed tokens for training, we overestimate the actual number of tokens effectively trained on by summing full context windows (in this case, 2048 tokens).

## D EXAMPLE PROOFS

Lean's `mathlib` is one of the most active open-source software projects in the world. More than one-third of the proofs found by our models are shorter and produce smaller proof terms than the ground truth, leading to dozens of GPT-f co-authored commits to `mathlib`. We examine some of the proofs found by our models in more detail.

### D.1 LIE_ALGEBRA.MORPHISM.MAP_BOT_IFF

This proof produces a proof term which is 4X smaller than the original:

```
lemma map_bot_iff : I.map f = ⊥ ↔ I ≤ f.ker :=
by { rw ← le_bot_iff, apply lie_ideal.map_le_iff_le_comap }
```

The original, human-written proof is much longer, viz.

```
lemma map_bot_iff : I.map f = ⊥ ↔ I ≤ f.ker :=
begin
  rw le_ker_iff, unfold lie_ideal.map, split; intros h,
  { rwa [eq_bot_iff, lie_submodule.lie_span_le, set.image_subset_iff,
    lie_submodule.bot_coe] at h,},
  { suffices : f '' I = ↑(⊥ : lie_ideal R L'), { rw [this,
    lie_submodule.lie_span_eq], },
    ext x, rw [lie_submodule.bot_coe, set.mem_singleton_iff,
    set.mem_image],
    split,
    { rintros ⟨y, hy, hx⟩, rw ← hx, exact h y hy, },
    { intros hx, use 0, simp [hx], }, },
end
```

### D.2 PRIMREC.OF_EQUIV

This proof produces a proof term which is 12X smaller than the original:

```
theorem of_equiv {β} {e : β ≃ α} :
  by haveI := primcodable.of_equiv α e; exact
  primrec e :=
by letI : primcodable β := primcodable.of_equiv α e; exact encode_iff.1
    primrec.encode
```

The author of the original proof and maintainer of that package commented:

> `encode_iff.1 primrec.encode` is clever, it's a way to translate `primrec`
> across an equivalence when the encode function is defined as `encode x =`
> `encode (e x)` where `e` is the isomorphism.

As far as they knew, this trick was never used before in the `computability` package.

### D.3 REAL.TAN_EQ_SIN_DIV_COS

This proof demonstrates our model's library knowledge and ability at premise selection.

```
lemma real.tan_eq_sin_div_cos (x : ℝ) : tan x = sin x / cos x :=
begin
  rw ← of_real_inj,
  simp only [complex.tan_eq_sin_div_cos, of_real_sin, of_real_cos,
    of_real_div, of_real_tan]
end
```

Our model was able to predict this entire list of `simp` lemmas in one shot. Note that the lemma `complex.tan_eq_sin_div_cos` in this list is the *complex number* version of the result, i.e. $\forall$ `(x : ℂ), tan x = sin x / cos x`. The previous human-written version of the proof did not use the more general version of the lemma on complex numbers, demonstrating our model's ability to find more general cases of lemmas. We contrast this with the human-written ground truth, which is more complex and performs a case analysis using the complex cosine:

```
  lemma tan_eq_sin_div_cos : tan x = sin x / cos x :=
if h : complex.cos x = 0 then by simp [sin, cos, tan, *, complex.tan,
    div_eq_mul_inv] at *
else
  by rw [sin, cos, tan, complex.tan, ← of_real_inj, div_eq_mul_inv,
    mul_re];
  simp [norm_sq, (div_div_eq_div_mul _ _ _).symm, div_self h]; refl
```

### D.4 SYM2.IS_DIAG_IFF_PROJ_EQ

The proof of this lemma is longer than the ground truth and was not contributed to `mathlib`, but we describe it here because the proof is original and includes a nontrivial instantiation of an existential quantifier.

```
theorem sym2.is_diag_iff_proj_eq (z : α × α) :
  is_diag ⟦z⟧ ↔ z.1 = z.2 :=
begin
    intros,
    simp only [is_diag, prod.ext_iff, quot.exists_rep, iff_true,
    not_true, eq_self_iff_true],
    simp [diag], split,
    { rintros ⟨y, hy⟩, cases hy; refl },
    intro h, cases z, existsi z_snd,
    cases h, refl,
end
```

Before `existsi z_snd`, the goal state is

```
z_fst z_snd: α
h: (z_fst, z_snd).fst = (z_fst, z_snd).snd
⊢ ∃ (y : α), (y, y) ≈ (z_fst, z_snd)
```

This goal state never appeared in `mathlib`.

### D.5  NORM_LE_ZERO_IFF

The following proof is remarkable because it uses fewer tactic steps and takes a different route to the proof than the ground truth, uses a complex idiom `simpa [...] using @...`, and was predicted in one shot.

```
lemma norm_le_zero_iff {α : Type u_1} [_inst_1 : normed_group α]
  {g : α} : ‖g‖ ≤ 0 ↔ g = 0 :=
by { simpa [le_antisymm_iff, norm_nonneg] using @norm_eq_zero α _ g }
-- ground truth:
-- by { rw[←dist_zero_right],
--      exact dist_le_zero }
```

The lemmas supplied between the square brackets are used to simplify the main goal. The lemma supplied after the keyword `using` can further simplify the lemmas supplied between the square brackets. The `@` modifier makes all arguments explicit. The string `@norm_eq_zero` never appeared in our training data but the prediction includes the correct number of correctly typed arguments, and even replaces the second argument with a placeholder _, correctly guessing that it can be inferred by the elaborator. Finally, this again showcases the strength of our models as premise selectors: all three lemmas `le_antisymm_iff`, `norm_nonneg`, and `norm_eq_zero` were not used in the human-supplied proof but are necessary for this proof.

Moving forward, we hope that our neural theorem provers will continue to find ways to improve `mathlib` and assist in creating new proofs. More generally, we hope neural theorem proving will one day be become a routine part of the formalization workflow.

