# OpenReview forum: "Proof Artifact Co-Training for Theorem Proving with Language Models"
_ICLR.cc/2022/Conference — ICLR 2022 Poster_

### Official Review · Reviewer_TVGW · 2021-11-02

**Correctness:** 4
**Technical Novelty And Significance:** 2
**Empirical Novelty And Significance:** 3
**Recommendation:** 5
**Confidence:** 3

**Main Review:**

This paper is an interesting application of a data augmentation or self-supervised learning type of approach for tactic based theorem proving. The idea itself is not completely new as the authors readily explain in the paragraph MACHINE LEARNING WITH PROOF ARTIFACTS on page 2. The paper appraisal therefore rests on the clarity of presentation, how convincing the experiments are, and how reproducible.

Overall the paper presentation is okay, although the clarity could be improved. One issue is that the main contribution is  mostly condensed into section 3.2 which is less than one page. In general the writing does not make the mechanisms by which proof artifacts may be extracted from Lean clear enough. It would be nice to include (possibly as an appendix) a focussed primer on Lean which allows to give more depth in the main paper while still being relatively self contained.

The experments are fairly convincing, although it is not entirely surprising that this approach works, and to repeat, the basic idea of extracting additional training data in this way is not entirely new.

TacticZero by Wu et al 2021 is missing from the related work. This is relevant because, by training end to end, that work effectively generates arbitrary amounts of training data through interaction the the HOL4 ITP system (intermediate theorems which are proven give some reward in that work).

Typos:
- synthesis -> synthesise
- DeepHOLZero -> DeepHOL
- wrong bold number in Figure 3


**Summary Of The Paper:**

This paper addresses the general setup of using transformer models for interactive theorem proving (ITP) tasks. The ITP engine considered here is Lean. The contribution of the paper is a data augmentation method. This is achieved by mining low level artifacts from a given dataset of Lean proofs. This includes data extracted from additional type information inferred by Lean while processing the given proofs. Several prediction tasks are formed for this additional data, to allow pre-training and co-training in combination with the existing WebMath dataset. Results show that the additional datasets extracted in this way aid substantively when used for pretraining and cotraining.


**Summary Of The Review:**

This paper gives a useful demonstration of how an existing idea (extract additional training data from the Lean engine) can be scaled up to give a tangible benefit in practice (improved theorem proving results).

---

> ### Author Response · Authors · 2021-11-21
> **Response to Reviewer TVGW**
>
> Thank you for your review.
>
> > This paper is an interesting application of a data augmentation or self-supervised learning type of approach for tactic based theorem proving. The idea itself is not completely new as the authors readily explain in the paragraph MACHINE LEARNING WITH PROOF ARTIFACTS on page 2. The paper appraisal therefore rests on the clarity of presentation, how convincing the experiments are, and how reproducible.
>
> While there is previous work which uses low level proof terms, that work mostly involves finding additional lemmas to use as premises. This is independent from and complementary to our method, which utilizes proof terms in a novel way to produce auxiliary training data for a generative language model which does not perform premise selection. Additionally, while reinforcement learning strategies like TacticZero allow one to extract more data from a subpart of a proof, this additional training data is still for the same task (e.g. predicting tactic applications) rather than for qualitatively different tasks like the ones used for PACT.
>
> The novelty of our work is that we present a method for extracting a diverse suite of auxiliary tasks (including premise selection as one task out of nine) from low-level proof terms which effectively transfer to the high-level tactic prediction objective. Our work is complementary to RL methods like that used in TacticZero, and PACT can be used to further augment the data from trajectories gathered from an RL policy.
>
>
> > Overall the paper presentation is okay, although the clarity could be improved. One issue is that the main contribution is mostly condensed into section 3.2 which is less than one page. In general the writing does not make the mechanisms by which proof artifacts may be extracted from Lean clear enough. It would be nice to include (possibly as an appendix) a focussed primer on Lean which allows to give more depth in the main paper while still being relatively self contained.
>
> We added some additional information to the appendix as suggested.
>
>
> > TacticZero by Wu et al 2021 is missing from the related work. This is relevant because, by training end to end, that work effectively generates arbitrary amounts of training data through interaction the the HOL4 ITP system (intermediate theorems which are proven give some reward in that work).
>
> Added.
>
>
> > Typos:
> > synthesis -> synthesise
> > DeepHOLZero -> DeepHOL
> > wrong bold number in Figure 3
>
> Fixed.  Thanks!

---

### Official Review · Reviewer_s8q7 · 2021-11-02

**Correctness:** 4
**Technical Novelty And Significance:** 2
**Empirical Novelty And Significance:** 3
**Recommendation:** 8
**Confidence:** 5

**Main Review:**

The main idea of using proof terms for self-supervised training is reasonable. The paper is well-written and the main approach is easy to understand. The experiment details are thoroughly demonstrated and the results look solid.

The main drawback of this paper is the lack of technique novelty in terms of matching learning.

Question: when extracting subterms from each proof term, do you parse a proof term according to the steps of human-written proofs, or do you parse the proof term according to its own grammar?

**Summary Of The Paper:**

This paper proposes to train large Transformers for proving theorems on Lean by using the auxiliary training objective built with proof terms. The main contribution of this paper is (1) building the theorem proving benchmark for Lean (2) building a sequence of auxiliary training tasks using proof terms and verifying the effectiveness (3) comparing the pre-training method with the co-training method.

**Summary Of The Review:**

Overall, this paper advances the techniques of training deep networks for theorem proving by co-training on proof terms. I recommend accepting this paper.

==============================
After reading the authors' responses and other reviewers' comments, I maintain my previous rating of this paper.

---

> ### Author Response · Authors · 2021-11-21
> **Response to Reviewer s8q7**
>
> Thank you for your review.
>
>
> > Question: when extracting subterms from each proof term, do you parse a proof term according to the steps of human-written proofs, or do you parse the proof term according to its own grammar?
>
> The proof terms are parsed according to their own grammar and we extract data from every subtree. Some subtrees correspond to certain steps of human-written proofs, but our data extraction procedure does not recognize this distinction, so we gather strictly more data than if we parsed proof terms according to the steps of human-written proofs.

---

### Official Review · Reviewer_K4ag · 2021-11-02

**Correctness:** 2
**Technical Novelty And Significance:** 2
**Empirical Novelty And Significance:** 2
**Recommendation:** 5
**Confidence:** 4

**Main Review:**

Below is my (mildly edited) previous review for another conference that the authors have seen more than half a year ago.

There are no significant changes to the major issues of the paper and hence practically no changes to my review.

The major issues with this work remain to be:

1. Poor data hygiene. Training on an unpublished WebMath corpus that may contain unknown amount of information about the testing data. This practically invalidates most of the experimental results. Training on (html-ized/isomorphic version of) the testing data is as much taboo in ML as circular proofs (proving T by T) in math. Unless there is full disclosure, there is no way to assess the results and compare them to other results obtained by established ML/ATP training/evaluation methods. If nothing else, the authors had more than half a year to publish their training dataset and let the readers and reviewers analyze it.

2. No ATP evaluation of the system when trained without WebMath. Given (1) this should have been done long ago but is still missing.

3. To re-iterate on (1) and the futility of not addressing the issues head-on, note that the checks for contamination (p. 14) added in this version of the paper are again poorly designed. A quick search of the html code of mathlib on the web shows pages such as https://leanprover-community.github.io/mathlib_docs/order/zorn.html where the rintro keyword is wrapped in the ">" "<" tokens (likely to do syntax highlighting in html).

Old review:

%%%%%%%%%%%%%%%%%%%

Strengths:

- The largest contribution is that this nontrivial engineering work has
been done for the first time for Lean, a relatively young but quickly
developing proof assistant. Making a working system such as this is
never easy, due to many technical issues. The system works and already
assists Lean users.

- The results seem encouraging, reaching between 32% and 48% depending
on what is used for pretraining.

- The most honest evaluation on the temporarily held-out set gives 37%,
compared to 22% done by a rather naive search procedure. This is a
reasonable improvement.

- A number of training tasks are experimented with in the
Lean setting. Neither the tasks nor the results seem extremely novel
or surprising compared with systems like TacticToe and its successors,
but this is certainly solid and useful work done for Lean.


Weaknesses:

- Similar to the previous work on Metamath, it is hard to understand
what are the possible leaks between the pretraining "WebMath" corpus
and the testing set. This seems quite negligent when training large
models with almost a billion of parameters with a large capacity for
memorization. The evaluation should really include the success rate
without the WebMath pretraining.

- To expand on this, many ITP corpora are published on the web and in
public repos in many different forms, sometimes after various
syntactic translations (to latex, etc.), which are however often
easy to recover by neural architectures
(https://doi.org/10.1007/978-3-319-96812-4_22). It is completely
unclear to me how big would be the effect of GPT pretraining on the
test dataset translated in various ways. See e.g. the work of
Gauthier for relatively simple statistical methods that quite
reliably transfer the proof knowledge between syntactically
different corpora (https://doi.org/10.1016/j.jsc.2018.04.005 ,
https://doi.org/10.1007/978-3-662-48899-7_26 ).

- It is unclear what resources go into the test evaluation that
remotely uses gptf. This makes an honest comparison with non-remote
methods using standard hardware difficult. This should be at least discussed.

- I do not understand the argument that RL-based data synthesis is
more expensive. The RL-style proof data synthesis done in TacticToe,
http://arxiv.org/abs/1805.07563 or
https://doi.org/10.4230/LIPIcs.ITP.2019.34 is most likely orders of
magnitude cheaper than just the pretraining done on WebMath here.

- While the contribution is solid and data augmentation methods are certainly useful in DL, the introductory claims about data scarcity being a newly encountered difficult problem in ML-for-TP are uninformed.
In particular, all of the larger ITP corpora (Isabelle, Mizar, HOl, Coq) are capable of easily exporting
millions of problems and proofs to start with and this has been done
many times since long ago. Already the 2003 version of MPTP and
the AI/TP experiments based on it
(https://doi.org/10.1007/s10817-004-6245-1) allowed and announced a
straightforward generation of 630000 related proof tasks from
Mizar. Further millions/billions/zillions of proving and training
tasks can be created easily by chasing the large derivation graphs
of the ITP libraries. This has been to various extent used in works
such as [1,2,3], where the benefit of learning from additional proof
tasks was also demonstrated. Equally easy and cheap is chasing the
graphs of large ATP proofs and generating problems from them,
running ATPs to generate terabytes of further data, etc. The theorem
proving domain is really the antithesis of data scarcity, has been
such for long time, and it is one of its great advantages over NLP
domains.

- A number of related experiments have been done with argument,
witness, conjecture and proof (step) synthesis recently. See e.g. [4-7].

[1] Cezary Kaliszyk, Josef Urban:
Learning-assisted theorem proving with millions of lemmas. J. Symb. Comput. 69: 109-128 (2015)

[2] Kaliszyk, C., Urban, J. & Vyskocil, J. Lemmatization for Stronger Reasoning in Large Theories in
FroCoS 2015 9322 (Springer, 2015), 341–356.

[3] Bartosz Piotrowski, Josef Urban:
Stateful Premise Selection by Recurrent Neural Networks. LPAR 2020: 409-422

[4] Thibault Gauthier:
Deep Reinforcement Learning for Synthesizing Functions in Higher-Order Logic. LPAR 2020: 230-248

[5] Thibault Gauthier:
Deep Reinforcement Learning in HOL4. CoRR abs/1910.11797 (2019)

[6] Bartosz Piotrowski, Josef Urban:
Guiding Inferences in Connection Tableau by Recurrent Neural Networks. CICM 2020: 309-314

[7] Josef Urban, Jan Jakubuv:
First Neural Conjecturing Datasets and Experiments. CICM 2020: 315-323

UPDATE:

I do not agree with the idea that the potential test set contamination would be a significant advantage due to its autoformalization potential:
- The Wang et all 2018 paper I mentioned and their related 2020 paper show that RNNs and Transformers are very good at such translations. I don't agree this would be surprising today for GPT.
- This is really an opposite of a "significant advantage". The test set evaluation is thus made incomparable to any other honest evaluation done on Lean in the future.
- The WebMath dataset has not been published and is impossible to check by readers and researchers. I would expect at least its publication as a partial response to such concerns.
- Not doing the test set ATP evaluation without the WebMath pretraining is a serious omission.

I also do not see any improvement in explaining the hardware resources used for the evaluation. This again makes the numbers here hard to compare with for other researchers and methods.

There is also hardly any improvement of the overclaims (noted also by other reviews) about the technical novelty, claims about comparable slowness of RL setups, etc.

%%%%%%%%%%%%%%%%%%%



**Summary Of The Paper:**

The paper describes a transformer-based tactical prover for Lean and its training methods.
The work is useful and the results are encouraging, however, there are also major issues.


**Summary Of The Review:**

The work is an important practical step for Lean, but there are major evaluation and presentation issues.

---

> ### Author Response · Authors · 2021-11-21
> **Response to Reviewer K4ag**
>
> We would like to thank the reviewer for their continued engagement with us and this work.
>
> The main concern of the reviewer is with possible direct contamination where the proof of a test theorem is directly in the pre-training data.  We went back to our data and searched for all the theorem identifiers (in their shortest form) that appear in any of the pre-training data, both the WebMath and the English language pre-training data.  We found the occasional identifiers, and in a few cases, a Lean file with a proof of a test theorem.  We show that after removing any potentially contaminated test theorems, our recalculated pass-rates only deviate by up to 1 percentage point and our main results continue to hold.
>
> We also added some discussion to the appendix where we look at the pass-rate when restricting to theorems which were added after the pre-trained data was collected.
>
> We also added the pass-rate for one run in Table 3 without WebMath training (using only English-language pre-training), but due to technical constraints are unable to provide the other such pass-rates.  However, we do provide validation losses for all such runs which closely correlate with the pass-rate throughout.
>
> Our main contribution in the paper is the effect of adding in the nine auxiliary tasks.  The effect is clearly visible in both the WebMath-pre-trained and non-WebMath-pre-trained versions, which makes us believe our contribution is largely independent of pre-training.
>
> The next main concern of the reviewer is that they feel our work is not comparable to other researchers, especially those with more limited resources.  It is common in machine learning for researchers to test the limits of what is possible through large models and extra data.  The fact that WebMath pre-training makes a significant difference is interesting to some readers including one of the other reviewers.  On the other hand, our model size and use of additional data should be taken into account in any comparisons.  We also provide benchmarks using smaller models which researchers can also compare against.  Moreover, decoder-only Transformer models similar to ours, even ones pre-trained on data very similar to WebMath, are publicly available to researchers in a variety of model sizes.
>
> There are also some remaining concerns that we address individually:
>
> > I do not understand the argument that RL-based data synthesis is more expensive.
>
> A small seed dataset of existing proofs could be used to initialize a policy for RL. However, additional compute must be exchanged for data by (1) using the policy to collect trajectories; (2) retraining the policy; (3) iterating the process, potentially many times to find suitable hyperparameters. What if you could extract an order-of-magnitude more auxiliary data from the same small seed dataset, in a single pass and without having to do any additional training or inference? PACT accomplishes this, and we show in our paper that jointly training on this auxiliary data produces a much better initial policy for theorem proving than otherwise. PACT can also be applied to any new proofs found throughout the course of an RL loop, and in this way we think of PACT as being complementary to rather than as a replacement for RL.
>
> > While the contribution is solid and data augmentation methods are certainly useful in DL, the introductory claims about data scarcity being a newly encountered difficult problem in ML-for-TP are uninformed.
>
> We believe this is a misunderstanding of our framing. Our work is motivated by the understanding that data scarcity is simply an important problem when applying machine learning to interactive proof assistant libraries, and all the more salient when using large language models, which are an attractive option due to their ability to generate (rather than select) premises but which require ample data to achieve predictable scaling of their performance.
>
> > It is unclear what resources go into the test evaluation that remotely uses gptf. This makes an honest comparison with non-remote methods using standard hardware difficult. This should be at least discussed.
>
> The resource numbers for training and testing can be found at the end of Appendix B. For each evaluation loop over the `test` set, we distributed the theorems over a pool of $32$ CPU workers whose inference requests were load-balanced over $4$ `V100` GPUs. Each evaluation required $\approx 10$ hours with $\approx 30\%$ GPU utilization.

---

### Official Review · Reviewer_Hmxh · 2021-11-03

**Correctness:** 4
**Technical Novelty And Significance:** 3
**Empirical Novelty And Significance:** 4
**Recommendation:** 8
**Confidence:** 4

**Main Review:**

* This work makes the following contributions.

  1. The PACT methodology: The paper proposes a methodology for extracting auxiliary tasks that can be trained jointly along with the main task (tactic prediction task). The research shows how low-level proof artifact data may be used to significantly boost performance on high-level theorem proving by co-training auxiliary tasks.
  The auxiliary tasks themselves will be useful for designing similar tasks for other theorem provers. More importantly the PACT methodology is more general; the idea of incorporating diverse auxiliary tasks as a language modeling task by introducing a distinct token for each task is novel. In my opinion, this idea has a potential to be applied to domains other than theorem proving.

  2. The paper contributes LEANSTEP dataset and the Learning environment. This is the first such dataset for the Lean Theorem prover. It is nontrivial to gather the data as it involves hooking into the Lean's compilation process. The source for generating the data is a big contribution to the theorem prover and machine learning community.

* The paper is well motivated and easy to understand and follow. The methodology is explained clearly and experiments are executed with a considerable amount of detail. I feel that the conclusions are in general well supported by the results.

* It is hypothesized that PACT acts a regularizer while imparting useful knowledge to the model due to mutual information across tasks. An ablation study is carried out to rule out the possibility that the benefits from PACT come from simply regularizing the model.

* It is interesting to know that WebMath pre-training is still helpful even in the presence of PACT. There are several such insights in the experiments section that will be helpful to the community.

**Weakness/Suggestions/Questions**

* Although The paper is well written overall, I think section 3.2 Proof Artifact Training can be improved by adding an example explaining the Lean terminology proof term, proof type, tactic, tactic state, etc. It is a bit difficult to understand the task definitions without first understanding the various constituents of the proof.

* Description of refl and tidy-bfs baselines appears much late in the paper. It would be nice if these baselines are described before Fig. 2 is referred.

* It is explained in the paper that the runtime environment ensures that the proofs are never circular. Is any care taken to handle this in training data?

* I am wondering why the authors chose Lean for this this work (as opposed to say MetaMath). Is it just because of the popularity of Lean or the richness of Lean tactics. Can PACT be applied to simpler theorem provers like MetaMath where there are no tactics? I suppose it would be hard to define auxiliary tasks for simpler systems. Would like to know author's view on this.

* page 3: s/"It has also been previous observed"/"It has also been previously observed"

**Summary Of The Paper:**

To deal with data scarcity in context of learning large Transformer language models for theorem proving, this work proposes a methodology (called PACT) for extracting auxiliary task data for joint training alongside the tactic prediction objective. The methodology has been applied to Lean proof assistant. PACT significantly improves the theorem proving success rate on held-out suite of test theorems.


**Summary Of The Review:**

Overall, I think PACT and the LEANSTEP dataset are significant contributions. The claim that PACT significantly improves the theorem proving success rate has been well supported by well designed experiments and ablation studies. The system has already contributed theorems to Lean library and has a potential of significant impact on the Lean community in future. I recommend acceptance of the paper.

---

> ### Author Response · Authors · 2021-11-21
> **Response to Reviewer Hmxh**
>
> # Response to Reviewer Hmxh
>
> Thank for your review.
>
> > Although The paper is well written overall, I think section 3.2 Proof Artifact Training can be improved by adding an example explaining the Lean terminology proof term, proof type, tactic, tactic state, etc. It is a bit difficult to understand the task definitions without first understanding the various constituents of the proof.
>
> We added some additional background in the appendix.
>
> > Description of refl and tidy-bfs baselines appears much late in the paper. It would be nice if these baselines are described before Fig. 2 is referred.
>
> Thanks.  We now introduce them earlier.
>
>
> > It is explained in the paper that the runtime environment ensures that the proofs are never circular. Is any care taken to handle this in training data?
>
> For our main experiments, we randomly partition the theorems into testing, validation, and training splits without accounting for the dependency graph between theorems. While this is a standard approach in the literature (e.g. HOList, Holophrasm) we also report strong evaluation results on the `future-mathlib` test set, which is guaranteed to avoid circular dependencies.
>
>
> > I am wondering why the authors chose Lean for this this work (as opposed to say MetaMath). Is it just because of the popularity of Lean or the richness of Lean tactics. Can PACT be applied to simpler theorem provers like MetaMath where there are no tactics? I suppose it would be hard to define auxiliary tasks for simpler systems. Would like to know author's view on this.
>
> The PACT data augmentation methodology can be applied to any theorem prover which represents proofs as trees; in particular it can be applied to Metamath, whose reverse Polish notation (RPN) proof terms implicitly define proof trees. All auxiliary tasks from the paper except for tasks 5 and 6 (involving elaboration, which involves automation that does not exist in Metamath) carry over directly.
>
> We chose Lean because of its popularity, the richness of its tactic language, and because its metaprogramming capabilites make it easy to inspect proof terms and extract the PACT data. Although the existence of a high-level tactic language does not affect whether or not one can extract PACT data, we consider the finding that jointly training on the low-level PACT data effectively transfers to high-level tactic theorem proving to be a key result of our work.
>
>
> > page 3: s/"It has also been previous observed"/"It has also been previously observed"
>
> Fixed.  Thanks!

---

### Author Response · Authors · 2021-11-21
**Response to all reviewers**

We would like to thank all the reviewers for taking the time to review our paper and for all their comments.  Before responding to each review individually, we would like to address the group.

We thank the reviewers for acknowledging the novel aspects of the work, including comments that **PACT and the LEANSTEP dataset are significant contributions**, **the results seem encouraging**, **the experiments are fairly convincing**, and **it is interesting to know that WebMath pre-training is still helpful even in the presence of PACT**.

We made the following changes to the paper:
* Added additional background to the appendix on Lean terms, tactics, tactic states, etc. with examples to better complement Section 3.2.
* Expanded our discussion in the appendix on pre-training data, including a re-assessment of pre-training data contamination and its impact on our results.
* Added a pass-rate for one of the runs without WebMath, and clarified some details about the ablation removing WebMath.
* (minor) Added extra citations.
* (minor) Fixed typos and made other small corrections.

---

### Decision · Program_Chairs · 2022-01-20

**Decision:**

Accept (Poster)

**Comment:**

This paper has potential impact in the theorem proving community, and demonstrated the possibility of using LMs for theorem proving in Lean, and is good enough to use "in the real world" through an interactive theorem proving tool.
The reviewers wish their data/models were public to address some concerns raised by the reviewers, but we think the community can benefit from this work.